

**Coseismic fluid–rock interactions in the Beichuan-Yingxiu surface rupture zone**
**of the Mw 7.9 Wenchuan earthquake and its implication for the fault zone**
**transformation**
Yangyang Wang [a]∗, Xiaoqi Gao [a], Sijia Li [b], Shiyuan Wang [c], Deyang Shi [a,d], Weibing Shen [e]∗∗
*a The Key Laboratory of Crustal Dynamics, Institute of Crustal Dynamics, China Earthquake Administration, Beijing,*
*100085, China*
*b. Geological Exploration and Development Research Institute, Chuanqing Drilling Engineering Co., Ltd., CNPC,*
*Chengdu, Sichuan, 610500, China.*
*c. Sichuan Earthquake Agency, Chengdu, 610041,China*
*d. Institute of Geophysics, China Earthquake Administration, Beijing, 100081, China*
*e. MLR Key Laboratory of Isotope Geology, Institute of Geology, Chinese Academy of Geological Sciences, Beijing*
*100037, China*
*\* Corresponding Author*
*\*\* CO-corresponding Author*
*E-mail: wyy871217@126.com*
*Phone number: +8615117973405*
*Present address: Key Laboratory of Crustal Dynamics, Institute of Crustal Dynamics, China Earthquake*
*Administration, No. 1, Anningzhuang Road, Haidian District, Beijing, People's Republic of China, 100085*



## Abstract

Mechanism of fluids in modifying mineralogy and geochemistry of the fault zone and the role of rock-
fluid interaction in the faulting weakening is still debatable. Through analyzing mineralogical
compositions, major elements as well as micro-structural characteristics of outcrop samples including
wall rocks, low damage zone, high damage zone and oriented fault gouge samples from principal slip
zone gouges, mineralogical and geochemical variations of the fault-rocks is observed from Shaba
outcrop of Beichuan-Yingxiu surface rupture zone of the Mw 7.9 Wenchuan earthquake, China. The
element enrichment/depletion pattern of fault rock shows excellent consistency with the variation
pattern of minerals in terms of the notable feldspar alteration and decomposition, decarbonization,
coseismic illitization, and chloritization that occurs in the fault zone. The Isocon analysis indicates that
the overall mass loss amount of the Shaba fault zone is ranked as low damage zone < high damage
zone < fault gouge, while the mass removal within the fault gouge causes the greatest loss amount in
the centeral strong-deformation region. The mechanism of material loss and transformation in the fault
zone, analyzed by comprehensive study, is found to be complicated: 1) during the coseismic period,
the mechanical fracturing, the dehydration reaction and thermal pressurization are likely the main
factors; 2) during the postseismic period, infiltration by the postseismic hydrothermal fluids is the key
factor. Therefore, the coseismic mechanical fracturing, chemical reaction related to coseismic
frictional heating, and postseismic fuild-rock interaction are important factors to change and control
the material composition and the fault zone evolution.
**Keywords**: fault gouge; mass balance transfer; fluid–rock interactions; coseismic fault; Wenchuan
earthquake; China



## 1. Introduction

During the seismic cycle, fluid action is commonly present in the fault zone, which mainly includes thermal pressurization and fuild-rock interaction. Thermal pressurization refers to the thermal pressurization effect of the fluid caused by rapid frictional heating, which substantially weakens the effective normal stress acting on the fault surface and the friction between two fault planes, which affects dynamic fault weakening and propagation of earthquake rupture (Sibson et al., 1973, 1990; Andrews, 2002; Wibberley and Shimamoto, 2003; Rice, 2006; Hayman et al., 2006; Mishima, 2009; Moore et al., 2013). Fluid-rock interaction means that the coseismic frictional heating intensifies the process of the fluid-rock interaction, changes the mineral composition, which mainly includes the mineral alteration/decomposition and dehydration (deaeration) (Forster et al., 1991; Hickman et al., 1995; Chen et al., 2007; Kaneko et al., 2017), and generates a large amount of layered silicate minerals (such as clay) with relatively low friction coefficient (Wintsch et al., 1995; Vrolijk et al., 1999; Fu et al., 2008; Lockner et al.,2011), which weakens the fault. The fluid action within the fault zone affects the earthquake nucleation, dynamic rupture propagation, and postseismic fault healing (Brace and Byerlee , 1966; Sibson, 1973; Beach, 1976; Bruhn et al., 1990; McCaig, 1988; Forster et al., 2007; Rice, 2006; Caine et al., 1996; Evens et al., 1995; Faulkner et al., 2003; Ishikawa et al., 2008; Hamada et al., 2009; Paola et al., 2011), the study of which has important significance.

The fluid action within the fault zone is macroscopically represented as the mineral transformation and the zoning of different mineral types and is microscopically manifested as the stability, the gain and loss of elements, and the variation in isotopic compositions (Beck et al., 1992; Thordsen et al., 2005; Wiersberg and Erzinger, 2007; Pili et al., 2011). Previous studies of the fluid action within the fault zone have focused on the material transformation, the element migration and



mainly use geochemical approaches to trace the sources of fluids and analyze the infiltration and fuild-
rock interaction processes of fluids (Anderson et al., 1983; Evans et al., 1995; Goddard and Evans,
1995; Roland et al., 1996; Chen et al., 2007; Pili et al., 2002, 2011; Ishikawa et al., 2008; Chen et al.,
2013b; Duan et al., 2016; Kaneko et al., 2017). The geochemical characteristics of fault zones cutting
clastic sedimentary rocks differ from those cutting carbonate and magmatic rocks. However, the
coseismic presence of fluids within the fualt zones cutting clastic sedimentary rocks and the role of
fluid–rock interaction on the fault zone transformation still remain debates.

The 2008 Wenchuan Earthquake (Mw 7.9), which occurred in the Longmen Shan Fault System

(LFS) on the east margin of Qinghai-Tibet Plateau, China, had never occurred since the beginning of
the recorded history of the world and provides a natural experimental site for studying the fluid action
with the clastic sedimentary fault zone (Ran et al., 2013; Yang et al., 2012, 2013, 2014; Yao et al.,
2013; Zhang et al., 2014). The existing researches on exposures of the 2008 Wenchuan Earthquake
rupture indicated that the material composition and the fault zone evolution were formed by the
multistage superposition of seismic cycles (Chen et al., 2013b; Yang et al. , 2013, 2016; Duan et al.,
2016). However, most relevant studies focus on the cumulative effect of long-term interseismic fuild-
rock interaction, lacking of coseismic fluid-rock interaction. Besides, as the important migration
pathway and activity site of fluids within the fault zone, dense fractures and secondary faults are
commonly developed in the fault zones of the crust and distributed differently in various regions of
the fault zone. The differential distribution of pores and cracks across the fault zone, result in various
seepage channel types and fluids behaviors within different parts of the fault zone, which controls the
interaction between fluids and channels and further causes the notable variation in mineral components
and chemical compositions with time and space. These factors eventually affect the mechanical





properties and slip behaviors of faults. What are the changes in the rock mineral components and
geochemical characteristics within the fault zone respond to the Wenchuan Earthquake? What is the
difference of the mechanisms of material loss and transformation among the different regions across
the fault zone?

Against this background, we select Shaba (SB) outcrop in the northern section of the Beichuan -

Yingxiu coseismic rupture of the Wenchuan earthquake, where contains the maximum value of vertical
displacement and fresh fault gouge, as the study object. This paper reports changes in the mineralogical
and geochemical compositions across the fault zone through XRD、XRF and SEM-EDS at sizes of
several millimeters to centimeters, attempting to analyze the mineral transformation and element
migration in different regions of the fault zone at different scales during the coseismic period. In
addition, this paper calculates the mass loss and element mobility within the fault zone through the
Isocon, to analyse the fluid flow behavior and build material transfer patterns of fault zone cutting the
clastic sedimentary rocks, in order to further study coseismic fuild-rock interaction and the role of fluid
in the fault zone evolution.
**2. Geological setting**

The 2008 Wenchuan Earthquake, which occurred in the LFS on the east margin of Qinghai-Tibet

Plateau, producing the simultaneous ruptures of two faults (Beichuan-Yingxiu surface rupture zone
and Anxian-Guanxian surface rupture zone) (Xu et al., 2009; Fu et al., 2011;Yang et al. , 2014; Yao et
al., 2013) (Fig.1a, b). SB outcrop in the northern section of the Beichuan - Yingxiu coseismic rupture
of the Wenchuan earthquake (Fig.1b, c), where contains the maximum value of vertical displacement
and fresh fault gouge. The coseismic surface rupture zone across SB area generally shows a
northeastward trend, which mainly passes by the mountainside and is a continuously extending fault





escarpment (Ran et al., 2008; Shi et al., 2009; Yuan et al., 2013). The original inclination of the early
fracture in the fault zone in study area should be the same as the topographic slope, which are both
inclined to the northwest. Due to the influence of supergene gravity, the occurrence of the fault zone
in the 5-30 m segments near the ground surface is bent, reversed and countertilted to the southeast with
the characteristics of normal faulting and right translation, and displacements differ at different places.
In general, the vertical displacement ranges from 2.0 to 10.5 m, and the horizontal displacement ranges
from 2 to 10 m. In SB area, the maximum vertical displacement ranges from 11 to 12 m; the maximum
dextral horizontal displacement ranges from 12 to 15 m; and the maximum oblique slip displacement
ranges from 14 to 17 m. Because the near-surface fault plane tends to reverse, this outcrop is
morphologically manifested as a normal faulting strike-slip fault that tilts eastward and forms a
landform of a valley within the slope (Ran et al., 2008; Yuan et al., 2013). The locations of the hanging
wall and footwall of the fault are defined in terms of the surface manifestation of this fault (Ran et al.,
2008; Shi et al., 2009; Yuan et al., 2013). The exploratory trench of the sampling site is perpendicular
to the trending of the surface rupture zone and stretches over the hanging wall and footwall of the fault.
The fault zone at the trench has a trending of NE45°-60° and a dip angle of 55°-85°. The northwest
wall is lifted, and the oblique scratches in the direction of 190°-240°∠40° can be observed on the fault
surface. Figure 1b shows the rupture zone of the ground surface at the trench. The footwall of the fault
is the black shale of the third subgroup ($S_{2\text{-}3}$) of the Maoxian Group of the Upper and Middle Silurian,
and the hanging wall is the semicemented silty clay in the Quaternary Holocene yellow slope residual
diluvium ($Q_{4dl+pl}$). A layer of bluish gray fresh fault gouge exists between the hanging wall and the
footwall of the fault, which is distributed stably and continuously and represents the coseismic fault
gouge of the Wenchuan Earthquake.



### 3. Field methodology and laboratory analyses

### 3.1 Field methodology of key outcrop and sampling procedure

Field observations show that, centered on the principal slip surface (PSS), the fault zoning in the SB outcrop is obvious and includes, from the margin to the center, wall rocks, the damage zone (including high-damaged breccia zone and low-damaged cataclasite zone), and the fault gouge (Fig. 2). The fault gouge consists of very fine, sticky particles that are easily differentiated from the surrounding rock (Fig. 2a). To obtain the geochemical and mineralogical characteristics of fault rocks in the study area, the location of the fault gouge on the PSS is placed at point zero, and the fault rock is systematically sampled across the fault zone. To ensure the accuracy of the sampling, a surface layer with a thickness of approximately 0.3 m is removed to avoid interference from weathering. Positioned according to the distance from the fault gouge, samples are collected at a maximum spacing of 3 m in the damage zone, then collected more densely towards the PSS and the sampling spacing gradually decreases to a minimum of 0.1 m approached the PSS (Fig. 2).

### 3.2 Chemical and mineralogical experimental method

#### 3.2.1X-ray powder diffraction (XRD)

To identify the major and clay minerals for representative outcrop-derived samples, X-ray powder diffraction (XRD) analyses were conducted by Beijing Research Institute of Uranium Geology Analytical Laboratory. Measurements were made with a PANalytical X'Pert PRO X-ray diffractometer using Cu-Ka radiation under conditions of 40 mA and 40 kV. Diffraction patterns were obtained with 2θ range from 5° to 70° at the scanning speed of 1.0°/minute. The samples were ground below 2



μm grain size. The sample composition analysis is divided into two parts: rock-forming mineral
analysis and clay mineral analysis. Firstly, the clay was centrifugated from the sample and the total
clay content was calculated. Then non-clay minerals were directly deposited on glass slides for
diffraction analysis. For analyzing the nature of clay minerals, after centrifugation, saturated glycol
(EG) and high-temperature (550℃) glass slides were prepared for their XRD analyses to identify clay
minerals. After diffraction analysis, each mineral contents was finally calculated. The semi-
quantitative analysis of minerals is carried out on the software JADE according to the steps of
background deduction, smoothing, peak search and calculation. The content of each mineral in the
sample is calculated by the K value method, and the calculation formula is as follows:
$$C_i = \frac{I_i/R_i}{\sum_i^n I_i/R_i} \times 100\%$$

158   Where, $C_i$ is the content of test mineral i; $I_i$ is the diffraction intensity of the highest peak; n

is the number of mineral species in the sample; $R_i$ is the RIR value of test mineral, which is provided
by the PDF card of software JADE 6.5.
**3.2.2 XRF analyses**

162   The major elements of the samples were tested by the AxiosmAX X-ray fluorescence

spectrometer at the Beijing Research Institute of Uranium Geology Analytical Laboratory. Major
elements of the samples were tested according to the determination of the Part 28 of chemical analysis
method of silicate rocks (GB/T14506.28-2010); the ferrous oxide content were tested in accordance
with Part 14 (GB/ t14506.14-2010); the LOI was tested based on rock mineral analysis (4th Ed.) Part

16.20.



## 4. Results

### 4.1 Mineralogical results from XRD analyses

XRD mineralogical analyses were performed on the < 2 mm fraction of 15 samples across the SB
outcrop corresponding to the 2008 Wenchuan earthquake, sampled from fault gouge, damaged zones
and wall rocks. The major mineral assemblages and contents within the fault zone were recognized as
quartz, feldspar, calcite, pyrite, gypsum and clay minerals with no detectable smectite, while pyrite
and gypsum were not tested in gouge samples (Fig. 3). In general, except for individual samples ($Qtz_{SB-}$
$_{5-2}$%= 59.2% and $Qtz_{SB-2-6}$%= 63.5%), Qtz% of fault zone samples is relatively high and stable. Qtz%
of damage zones / wall rocks samples is about 35%, while Qtz% of fault gouge samples is slightly
lower, ranging from 30% to 35% basically. Potassium feldspar only exists in few samples, while
plagioclase is relatively developed with range of 9.9% -33.0% in damage zones / wall rocks and less
than 5% in fault gouge basically. The contents of carbonate minerals (calcite and dolomite) of fault
gouge are less than those in damage zones / wall rocks. On the contrary, clay minerals were
significantly developed in fault gouge, ranging from 46.2% to 62.0%, which was significantly higher
than that of the surrounding damage zones / wall rocks.
Because the mineralization of the fault and the matrix cannot be completely differentiated, no
significant difference between the mineral types of the wall rock and the damage zone is detected,
whereas the various mineral contents of the samples from the PSS to the damage zone are notably
different (Fig. 3). The mineral assemblage exhibits continuous variation from the damage zone to the
fault core: (1) the content of quartz and feldspar (potassium feldspar and plagioclase) declines
remarkably, and the feldspar content declines by approximately 30% and even decreases to 2.8% in





the fault gouge; (2) the content of carbonate minerals (calcite and dolomite) decreases and gradually
becomes zero in the fault gouge (or below the detection limitation); and (3) the total amount of clay
minerals increases dramatically and even increases to a maximum of 61% in the fault gouge (Fig. 3).

Note that more than 30% of the minerals in the fault zone are clay minerals, which mainly include

illite, chlorite, illite/smectite mixed-layer (I/S), and a small amount of kaolinite, and the smectite exists
in the form of I/S (except for sample SB-1-9, which has an S% of 2%). The content of illite and chlorite
in clay minerals, which are 39%-75% and 11%-34%, respectively, are the highest, followed by the
content of I/S (6%- 44%), and the I/S ratio (percentage of smectite in the I/S) almost remains between
5% and 7% (Fig. 3). A significant difference is observed between the clay minerals at various parts of
the fault zone: (1) the total content of illite in the fault gouge (i.e., illite contained in I*=I/S + illite) is
greater than that in the damage zone and wall rock, while the total content of smectite (S*= smectite +
smectite in I/S) is smaller than that in the damage zone and wall rock; (2) the content of chlorite in the
fault gouge is higher than that in the damage zone and wall rock, and the chlorite content tends to
gradually increase from the high damage zone to the fault core (Fig. 3).
**4.2 Elements results**

The analysis of the major elements of the samples from the SB outcrop in the Beichuan-Yingxiu

surface rupture zone shows that the contents of some elements are relatively stable, while the contents
of other elements vary greatly, for example, $Al_2O_3$ (11.86%-19.21%), $Fe_2O_3^T$(3.67%-7.57%), CaO
(1.19%-7.79%), $Na_2O$ (0.554%-3.97%), and $K_2O$ (1.54%-4.20%). Similar to the mineral composition
of rocks, part of the major elements exhibit the characteristics of differential distribution in the fault
zone: (1) the $Al_2O_3$, $Fe_2O_3^T$, and $K_2O$ contents become increasingly towards the fault gouge and exhibit



significant enrichment in the fault gouge; (2) conversely, the contents of $Na_2O$ and $P_2O_5$ gradually
decrease towards the fault gouge and exhibit significant depletion in the fault gouge. In addition, the
$SiO_2$ and CaO elements exhibit slight decrease towards the fault gouge, and the contents of MgO, MnO,
and $TiO_2$ elements remain unchanged (Table1 and Fig.4).

**5. Discussion**

**5.1 Mass balance transfer across the fault zone**

The differential distribution characteristics of mineral components and elemental composition of

the wall rock, damage zone, and fault gouge samples in the Beichuan-Yingxiu surface rupture zone
show that coseismic fracturing, which is a nonclosed (i.e., open) dynamic geological process, is
characterized by significant fuild-rock interaction, gain and loss of component/energy, and mass
balance transfer across the fault zone. To eliminate to some extent the influence of the variation in the
total amount of material in the nonclosed system, the element that remains relatively stable during the
geological process and whose quantities (basic quantities such as mass, volume, and density) do not
vary is employed as a baseline, and then the migration and variation of certain components during the
geological process are quantitatively compared. According to the variation trend of chemical elements
in the samples across the fault zone characterized by the bivariate diagram (Fig. 5), the $Al_2O_3$, $Fe_2O_3{}^T$,
$K_2O$, and FeO concentrations in the fault gouge is higher than that in the damage zone, while CaO and
$Na_2O$ concentrations shows the opposite trend. The concentration of $TiO_2$ is relatively stable (with the
maximum difference of only approximately 0.25%) and relatively high in the fault gouge, and $TiO_2$
has excellent correlation with other major elements (Fig. 5). Considering factors such as the activity,
geological process, element content and detection limitation, the oxide of the high field-strength



element (HFSE) Ti is selected in this study as the immobile component, and TiO₂ wt.% is used to
evaluate the relative migration rate of the rock during the alteration process in the fault zone, and the
mineral alteration and dissolution-precipitation process that occur in the fault rock during the coseismic
process are analyzed.

**5.1.1 Mass removal in the fault gouge**

Characteristics such as mineral types and contents, as well as chemical compositions of the fault

gouge can be considered as the response to the mineralogical and geochemical characteristics of the
fault gouge to the slip pattern and activity of the fault. A small-scale, dense sampling is conducted in
the outcropping fault gouge of SB to observe the mass removal state and geochemical transformation
mechanism in different regions of the fault gouge. According to the principle of mass balance, in the
open process of a geological system, the wall rock sample 'O' is transformed into sample 'A' after a
series of component migrations. The mass ($M_k^A$) of any component k in sample 'A' shall be equal to
the sum of the mass ($M_k^O$) and transfer mass ($\Delta M_k^{O-A}$) of the component k in sample 'O', namely:

$$M_k^A = \Delta M_k^{O-A} + M_k^O \quad (1)$$

Dividing throughout by $M^O$ to obtain:

$$\frac{M_k^A}{M^O} = \frac{\Delta M_k^{O-A}}{M^O} + \frac{M_k^O}{M^O} = \Delta C_k^{O-A} + C_k^O \quad (2)$$

Otherwise, $C_k^A = \frac{M_k^A}{M^A} = \frac{M_k^A}{M^A}\frac{M^O}{M^O} = \frac{M^O M_k^A}{M^A M^O} = \frac{M^O}{M^A}\left(\frac{\Delta M_k^{O-A}}{M^O} + \frac{M_k^O}{M^O}\right) = \frac{M^O}{M^A}\left(\Delta C_k^{O-A} + C_k^O\right) \quad (3) $ (Grant, 1986)

In the open process of the geological system, component Ti is selected as the immobile component,

that is, there is no increase or decrease in the mass of Ti in this process, which means $\Delta C_{Ti}^{O-A} = 0$,
from equation (3) we have:

$$C_{Ti}^A = \frac{M^O}{M^A} C_{Ti}^O \quad (4)$$

Therefore, this line, for which slope is equal to $\frac{M^O}{M^A}$, can be called an 'isocon', that is, a line





connecting points of equal geochemical concentrations. At the same time, the slope of Isocon $\frac{M^O}{M^A}$
actually defines the change% in the mass of the sample before and after the geological process.

By this method (Grant, 1986, 2005; Gresens, 1967; O'Hara and Blackburn, 1988, 1989), mass

loss rate (M%) and mass balance equations can be calculated/written for six sections of the fault gouge
on the basis of changes in element distribution. The M% for different sections of the fault gouge are
slightly different (<5%), and the following pattern exists: The M% reaches the maximum at the center
of the fault gouge and gradually decrease towards the gouge margin. According to the results from
previous studies of the microstructure of the fault gouge in the SB outcrop (Yuan et al., 2013), two
well-developed shear planes exist near the center of the fault gouge with their direction parallel to the
main shear direction of the fault. These belong to Y-shear, which is the principal slip surface of the
Wenchuan Earthquake fault. The deformation of the fault gouge in the region between two Y-shears
(central strong-deformation region) is more intense than that on the two sides, and the microstructural
characteristics of various typical deformations, including Riedel shears, P-foliation, P-shear, and
trailing structures, are developed in this region. Thus, the slip deformation is the most concentrated in
this part of the earthquake. The mentioned microstructure differences among different parts of the fault
gouge may explain the M% differential distribution across the fault gouge to some extent: the
coseismic rupture causes the reduction in grain size and increase in the fuild-to-rock rate, and more
stress concentrated in the centeral strong-deformation region, result in the strengthening of fuild-rock
interaction. The mass loss amounts are the largest in the central part, followed by those on the two
sides, and the amount of each element lost per 100 g of wall rocks is calculated (Table 2).





### 5.1.2 Mass migration of continuous multisample system in the fault zone

Overall, the components of the rock samples in different regions undergo various degrees of

transfer, that is, forming a series of samples with continuous variation in multiple components along

the fault zone, rather than just two distinct samples (unchanged sample and changed sample) (Mori et

al., 2007; Li et al., 2007; Beinlich et al., 2010; Guo et al. , 2009, 2013). In this study, the standardized

Isocon diagram method (Anormalization solution using Isocon diagram) (Guo et al., 2009) is employed

to analyze a series of samples across the fault zone of the SB outcrop.

Assume that 'O', 'A', and 'B' are a series of samples formed by progressive coseismic

geochemistry and geophysics alteration in the Beichuan - Yingxiu fault zone(Fig. 6). The relationship

between sample 'O', 'A' and 'B' are described as follows:

$$C_m^A = \frac{M^O}{M^A}\ (1+\Delta C_m^{O-A}/C_m^O)\ C_m^O;\ (5)$$

$$C_m^B = \frac{M^O}{M^B}\ (1+\Delta C_m^{O-B}/C_m^O)\ C_m^O\ (6)$$

The immobile component $T_i$ and mobile component $m$ in sample B were further standardized.

This standardization process provided a common Isocon line of component $m$ in each sample without

changing the M%.

The calculation using the above standardized Isocon method shows notable transfer

characteristics of the major elements in a series of samples formed during the earthquake process: the

major elements of the low damage zone are mostly concentrated near/on the isocon line, while those

of the fault gouge are mostly distributed on the two sides (Fig. 6a), which indicates the higher transfer

rate of elements in the fault gouge. Specifically, CaO, Na$_2$O, P$_2$O$_5$, SiO$_2$, LOI, and MnO are depleted

in the fault core. The transfer rates of CaO and Na$_2$O in the fault gouge samples are the largest, with

the average values of -82% and -89%, respectively (Fig. 6a). Conversely, FeO and K$_2$O are





significantly enriched in the fault core, with the average values of 42% and 71%, respectively, and
$Al_2O_3$ and $Fe_2O_3^T$ are slightly enriched.
According to the material balance equation and element transfer parameters, the fault gouge, high
damage zone, and low damage zone relative to the wall rock in the SB outcrop, the M% is mainly
ranked as low damage zone < high damage zone < fault gouge. The M% is relatively small in the low
damage zone, which indicates that the low damage zone mainly undergoes relatively small mechanical
fracturing and chemical alteration. In contrast, the M% in the high damage zone and fault gouge
gradually increase (Fig. 6a and Table 3), which indicates the loss of a relatively substantial amount of
material.

**5.2 Mineral and geochemical transformation during the seismic cycle**

**5.2.1 Decomposition and alteration of feldspar and depletion of Na and Si**

Towards the PSS of the study area, the feldspar content significantly decreases, while the total
clay content gradually increases (Fig. 3), which indicates that the dissolution and alteration of feldspar
minerals might occur in the fault core. The contents of $Na_2O$ are positively correlated with the contents
of feldspar minerals (Fig. 7a), which means the feldspar-related alkaline earth elements (e.g., $Na_2O$)
after feldspar dissolution were taken away by fluids, which causes the notable depletion of $Na_2O$ in
the fault gouge (Fig. 7a), with transfer rate of -89%, which also confirms the presence of feldspar
decomposition and alteration. In addition, the microscopic mineral identification shows that the
plagioclase in the fault gouge has apparently been altered to clay minerals (Fig. 9). Based on this
analysis, the formation of some neogenic clay minerals in the fault zone is related to the alteration of
feldspar, and the fluid-rock reaction may mainly include the alteration and transitions of plagioclase



to kaolinite and chlorite:
1) $2NaAlSi_3O_8$ (plagioclase) + $9H_2O + 2H^+ \rightarrow Al_2Si_2O_5(OH)_4$ (kaolinite) $+2Na^+ + 4H_4SiO_4$
2) $2NaAlSi_3O_8$ (plagioclase) $+4(Fe, Mg)^{2+} + 2(Fe, Al)^{3+} + 10H_2O \rightarrow (Mg, Fe)_4(Fe, Al)_2Si_2O_{10}(OH)_8$
(chlorite) $+ 4SiO_2 + 2Na^+ + 12H^+$
These reactions generate a large amount of $SiO_2$ component (Goddard and Evans, 1995; Arancibia
et al. 2014; Duan et al., 2016), which is dissolved in the fluid (Goddard and Evans, 1995), and the
resulting water-soluble $SiO_2$ undergoes transfer and loss during the process of seismic fault slip, which
causes significant depletion of the $SiO_2$ component in the fault core.
**5.2.2 Transition of smectite and illite in the I/S**
Illite and smectite (mainly in the form of I/S) are relatively enriched within the fault zone in the
SB outcrop; otherwise, their contents in the wall rock and low damage zone are significantly different
from those in the fault gouge and high damage zone. Previous studies have shown that part of illite
and I/S in the fault core have been proved to be neogenic clay minerals (Solum et al., 2005; Chen et
al., 2007). According to the $K_2O-Al_2O_3$ bivariate diagram (Fig. 8a), the illitization is found to be
widespread within the fault zone, which provides a basis for the formation of neogenic clay minerals.
The degree of illitization differs at different parts of the fault zone: the degree of illitization of the fault
gouge is significantly higher than that of the damage zone and wall rock; and the transition rate from
I/S to I is relatively high (Fig. 8b).
The enrichment of clay minerals and the illitization within the fault core may be controlled by the
frictional heating, which intensifies the process of fuild-rock interaction, accelerates the alteration and
decomposition of minerals, and implements dehydration (deaeration). Illite and smectite are the two



2-terminal minerals of I/S. During the coseismic frictional heating, part of the interlayer water is
extruded from the smectite and transforms part of I/S to illite (Ma and Shimamoto, 1995; Hirose and
Bystricky, 2007; Sulem and Famin, 2009; Lin et al., 2013). This progressive transition from smectite
to illite is smectite $+ K^+ + Al^{3+} \rightarrow 1$ illite $+ Na^+ + Ca^{2+} + Si^{4+} + Fe^{2+} + Mg^{2+} + H_2O$. In this reaction
process, the fault can be used as a dehydration channel for the transition of I/S to illite; the fractured
and altered minerals (Moore et al.1997), such as feldspar provide $K^+$ for the formation of illite, and the
water-soluble $SiO_2$ generated from the transition may migrate and be lost along the fault (Goddard and
Evans, 1995; Arancibia et al. 2014; Duan et al., 2016).
It is worth noting that the dehydration of smectite and kaolinite can be complete during coseismic
period, while the transition from smectite to illite needs more time, which may explain why the
illitization of the smectite in the fresh coseismic fault gouge of SB is limited to the slight transition in
the I/S layer (transition from smectite-rich I/S to illite-rich I/S). At the same time, smectite is generally
formed under alkaline conditions, and the medium-acid or acidic fluid environment of SB area(Chen
et al., 2013b; Duan et al., 2016)may inhibits its formation.
**5.2.3 Decarbonization and depletion of Ca**
The carbonate content in the fault gouge of the SB outcrop is lower than that of the wall rock and
damage zone, which reflects that decarbonization may occur in the fault rock during the coseismic
process. Note that the presence of decarbonization of the wall rock indicates that the decarbonization
range is wider than the distribution range of the fault gouge. In the process of dissolution and thermal
decomposition of carbonate minerals, the relevant elements are prone to be removed by fluids and
become depleted (Chen et al., 2013b). As the main compositional element of carbonate minerals (eg.





calcite and dolomite), the contents of CaO are positively correlated with the contents of carbonate
minerals (Fig. 7b), which means the Ca element after feldspar dissolution were taken away by fluids,
the content of CaO decreases towards the fault gouge, which indicates the decarbonization, the
decomposition and consumption of calcite gradually strengthen towards the fault gouge (Fig. 4). In
addition, the microscopic mineral identification shows that the carbonate in the fault gouge has been
altered to clay minerals (Fig. 9); the depletion of Ca reaches the maximum in the fault gouge with the
highest degree of coseismic effect; and the enrichment of Ca in very few fault gouge samples is likely
related to the inclusion of carbonate particles and calcite-rich veins. Other major components of
carbonate, such as MgO and LOI, also show a similar distribution (Fig. 4).
**5.2.4 Extensive chloritization**

Compared with other outcrops of the Beichuan-Yingxiu surface rupture zone, the fault gouge at

the SB sampling site is rich in chlorite (with a maximum content of 25%), and the chlorite content
tends to gradually increase towards the fault gouge. Microscopic mineral identification shows that
chlorite are substantially developed in the mineral surface / gain edges and rock pores (Fig. 9), and the
structure of feldspar altered to chlorite often occurs in the fault gouge samples (Fig. 9), which reflects
the extensive chloritization in the fault zone.

The mineral alteration and structural characteristics of some samples of the fault gouge and

damage zone are observed and analyzed by combined scanning electron micrography (SEM) and
energy dispersive X-ray spectroscopy (EDS), which also indicates notable chloritization in the samples.
The figure shows the altered mineral structure at the edge of feldspar particles in the grayish green
fault gouge. The chlorite of altered feldspar type is distributed on the surface of feldspar, around



feldspar, or in cracks (Fig. 9b, c, d, e and f). The EDS analysis of feldspar particles from the middle to
the edge shows that the components among feldspar particles are essentially unaffected, and the major
compositional elements are Si, Al, O, and K (Fig. 10a), which suggests a typical potassium feldspar.
From the middle to the edge, elements such as Fe and Mg gradually appear in the elemental
composition, and the closer to the edge are the elements, the higher are the contents of the Fe and Mg
elements (Fig. 10b), which are gradually consistent with the elemental composition of the surrounding
neogenic clay minerals, such as chlorite. The EDS line scanning of the clay minerals formed by the
alteration of feldspar particles also indicates that the newly formed minerals are characterized by
progressive enrichment of Fe and Mg elements (Fig. 11).
The formation of chlorite in the fault zone occurs two ways: 1) direct decomposition of mafic
silicate minerals, and 2) metasomatic alteration of Fe and Mg components caused by hydrothermal
solution. Mafic minerals are not developed in the study area, which imply large amounts of chlorite
are unlikely to be derived from the direct decomposition of mafic silicate minerals. Therefore, the
extensive chloritization may related to the fluid environment and the ion types of fluids in the study
area: the coseismic rupture causes the periodically and cyclically injected atmospheric precipitation to
continuously react with the wall rock. If $Mg^{2+}$ is added to the mentioned system, chlorite will be
generated. In the study area, the feldspar alteration and decarbonization occur in the fault zone,
especially the dissolution of carbonate may provide $Mg^{2+}$ for the fluid, and the dissolution of Fe-rich
dolomite could also provide $Fe^{2+}$. The fault zone in the study area inhibits the formation of smectite in
the acidic environment and promotes the chloritization of minerals. The reaction for the chloritization
and alteration of plagioclase in the fault gouge of the study area may be forms following the equation:
$2NaAlSi_3O_8 \text{ (plagioclase)} + 4(Fe, Mg)^{2+} + 2(Fe, Al)^{3+} + 10H_2O \rightarrow (Mg, Fe)_4(Fe,Al)_2Si_2O_{10}(OH)_8$


(chlorite) + $4SiO_2$ + $2Na^+$ + $12H^+$

**5.3 Fault zone transformation**

The mineral compositions and geochemical characteristics of the Beichuan-Yingxiu surface
rupture zone of the Wenchuan Earthquake vary with time and space, which cause a significant
difference in the mineral components, elemental compositions, and mass loss of rock samples among
different regions of the fault zone in the study area. In terms of time, the degree of the main reactions
(i.e., feldspar alteration, illitization, decarbonization, and chloritization) within the fault zone and the
mechanism of material loss and transformation are different between coseismic and postseismic
periods of the Wenchuan Earthquake. In terms of space, the degree of the main reactions and the
mechanisms of material loss and transformation are different across the fault zone including the fault
core and damage zone in the study area. These differences experienced temporally and spatially by the
fault zone affect the mechanical properties and the slip behavior of the fault.
In the coseismic period, the mechanical fracturing of the fault and the coseismic dehydration and
thermal pressurization during the coseismic friction heating are the main mechanisms that cause the
material loss and transformation within the fault zone. Compared with the damage zone, the fault
gouge of the fault core experiences stronger mechanical fracturing, coseismic dehydration, and thermal
pressurization, which causes greater material loss and transformation in the coseismic period but
relatively weaker postseismic fluid infiltration. The relatively stronger mechanical fracturing causes a
reduction in the grain size, which corresponds to a relatively high specific surface area and chemical
potential and promotes the mass loss of the fault gouge (Fig. 12). This study of Section 5.1.1. reveals
that the amount of mass loss reached the largest in the strong deformation region at the center of the





SB fault gouge, followed by the region on the two sides with a lesser loss, because the stress of the

fault gouge is more concentrated in the central region, which causes a more significant reduction in

grain size. The coseismic dehydration and thermal pressurization during the coseismic friction heating

are conducive to the accumulation of high pore pressure in the fault core, which causes material loss,

and the existence of notable decarbonization within the fault gouge supports this view. The coseismic

thermal pressurization and dehydration have an important role in promoting the fracture process,

especially near the surface, in the Wenchuan Earthquake. The near-surface displacement of the

northern segment of the fault, where the study area is located, is generally larger than the deep

displacement, which may be related to the abnormally high coseismic slip displacement and velocity

near the SB section (Chen et al., 2013b).

In the postseismic period, fluid infiltration is the main mechanism for material loss and

transformation. The postseismic fluid infiltration causes relatively stronger material loss and

transformation in the damage zone, while relatively weaker in the fault core. The cross-fault

permeability in north segment of the Beichuan-Yingxiu surface rupture zone exhibits a typical "dual

structure," which is shown in other fault zones and composed of a low-permeability core, a high-

permeability damage zone with fracture development, and microfracture-bearing wall rock, among

which the fresh fault gouge has the lowest permeability (Cain, et al., 1996; Billi, 2005; Chen et al.,

2013b). The LOI content in the fault gouge is significantly lower than that in the damage zone in the

SB area (Fig. 9 and table 1), which suggests that the fault gouge has a relatively low water content and

its fluid permeability is lower than that of the damage zone. The "dual structure" causes the

interseismic fluid action in the fault zone to be mostly confined to the high-permeability damage zone.

Thus, the fluid can easily migrate in parallel to the fault but does not easily flow perpendicular to the





fault (Cain, et al.,1996; Billi, 2005; Chen et al., 2013b). Note that the high-porosity, high-permeability
damage zone with tensile fractures can provide channels for the hydrothermal fluids and promote fuild-
rock interaction such as mineral alteration, especially in postseismic periods when the fault valve is
temporarily opened. For example, typical hydrothermal minerals, such as pyrite and gypsum, are
developed in the damage zone in the SB area, while no minerals or veins crystallized from fluid are
observed in the fault gouge (Fig. 12). It indicates that the damage zone, rather than the fault gouge, is
the main active zone for the postseismic hydrothermal fluid. Multistage calcite veins exist in the high
damage zones on both sides of the fault gouge, and the fine-grained calcite veins heal the fluid channel
by rapid crystallization.
**6. Conclusions**

(1) The major mineral assemblages and contents within the fault zone of the SB outcrop in the

Beichuan-Yingxiu surface rupture zone were recognized as quartz, feldspar, calcite, pyrite, gypsum
and clay minerals with no detectable smectite, while pyrite and gypsum were not tested in gouge
samples. The mineral assemblage exhibits continuous variation from the damage zone to the fault core:
1) the content of quartz and feldspar (potassium feldspar and plagioclase) declines remarkably, and the
feldspar content declines by approximately 30% and even decreases to 2.8% in the fault gouge; 2) the
content of carbonate minerals (calcite and dolomite) decreases and gradually becomes zero in the fault
gouge (or below the detection limitation); and 3) the total amount of clay minerals increases
dramatically and even increases to a maximum of 61% in the fault gouge.

(2) The major elements of the samples from the SB outcrop in the Beichuan-Yingxiu surface

rupture zone shows that the contents of some elements are relatively stable, while the contents of other





elements vary greatly, such as $Al_2O_3$ (11.86%-19.21%), $Fe_2O_3^T$ (3.67%-7.57%), CaO (1.19%-7.79%),
$Na_2O$ (0.554%-3.97%), $K_2O$ (1.54%-4.20%). Similar to the mineral composition of rocks, part of the
major elements exhibit the characteristics of differential distribution in the fault zone: 1) the $Al_2O_3$,
$Fe_2O_3^T$, $K_2O$ contents become increasingly towards the fault gouge and exhibit significant enrichment
in the fault gouge; 2) conversely, the contents of $Na_2O$ and $P_2O_5$ gradually decrease towards the fault
gouge and exhibit significant depletion in the fault gouge. In addition, the $SiO_2$ and CaO elements
exhibit slight decrease towards the fault gouge, and the contents of MgO, MnO, and $TiO_2$ elements
remain unchanged.

(3) The Isocon analysis indicates that significant fuild-rock interaction, gain and loss of

component/energy, and mass balance transfer were existed across the fault zone in the study area, and
M% varied in different regions of the fault zone: 1) Within the fault gouge, the M% reaches the
maximum in the centeral strong-deformation region and gradually decrease towards the gouge margin;
2) Overall, the mass loss amount of the SB fault zone is ranked as low damage zone < high damage
zone < fault gouge.

(4) The notable feldspar alteration and decomposition, decarbonization, coseismic illitization, and

chloritization that occur in the fault zone, which generates a large amount of clay minerals and the
depletion of highly active elements (e.g., Na, Si, and Ca) related to feldspar and carbonate rock, as
well as the enrichment of elements related to aluminosilicate minerals in the core of the fault. The
extensive chloritization in the fault zone mainly due to metasomatic alteration of Fe and Mg
components caused by hydrothermal solution.

(5) The mechanism of material loss in the fault zone, analysed by comprehensive study, is found

to be complicated: 1) during the coseismic period, the mechanical fracturing, the dehydration reaction





and thermal pressurization caused by coseismic frictional heating are likely the main factors that result
in the material loss and transformation, especially within the fault core, which is stronger than those
in the damage zones; 2) during the postseismic period, it concludes that infiltration by the postseismic
hydrothermal fluids mainly controlled the material loss and transformation. Due to the better
permeability than the fault core, the damage zone is conducive to hydrothermal upwelling, fuild-rock
interaction, and fracture healing.
**Data availability**
All data generated or analyzed during this study are included in this article.
**Author contributions**
Yangyang Wang designed and prepared the paper. Xiaoqi Gao carried out the experiment. Sijia Li,
Siyuan Wang and Deyang Shi participated in the analysis and discussion of the final results. Weibing
Shen supervised the preparation of the paper.
**Competing interests**
The authors declare that they have no conflict of interest.
**Acknowledgments**
This work was supported by the research grant from Institute of Crustal Dynamics, China
Earthquake Administration (No. ZDJ2019-02), the special project of fundamental scientific research
for the central-level public interest research institutes (No. ZDJ2017-27) from the Institute of Crustal
Dynamics, China Earthquake Administration, the special project of monitoring and prediction (No.

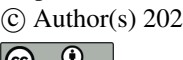



2018020212) from China Earthquake Administration, the "insight study on the magnitude 6.6
earthquake in Jianghe, Xinjiang" from the Institute of Earthquake Forecasting, China Earthquake
Administration.

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

**Table captions**
**Table 1**. Average major element composition of the fault gouge and the rocks from the high / low
damage zones and the SB outcrop.
**Table 2**. The material balance equation of samples within the fault gouge and the corresponding mass
loss rates (%).
**Table 3**. The material balance equation of samples across the fault zone and the corresponding mass
loss rates (%).
**Figure captions**
**Figure 1**. Maps showing the structural settings and location of the Shaba (SB) outcrop. (a) tectonic
setting of the Longmen Shan Fault System (LFS). Modified from Li et al. (2013a). (b) The district-
scale map showing the lithology and main branches of the Yingxiu-Beichuan fault around SB area. (c)
Regional distribution of Sichuan Basin and magnitude distribution in China.
**Figure 2**. SB outcrop: general observation. (a) Photograph of the SB outcrop with the coseismic fault
gouge and location of the sampling sites. (b) Fault gouge sampled using metal tubes (the gouge was
sampled, consolidated, and then cut to prepare thin sections of gouge sample after consolidation). (c)



Geological map of the SB outcrop.
**Figure 3.** Variation in major and clay mineral contents across the SB outcrop. Different shades of
yellow, from dark to light, represents the fault gouge, the high damage zone, and the low damage zone,
respectively. The black dashed lines represent the varying trends.
**Figure 4**. Variation in major elements contents across the SB outcrop. Different shades of yellow, from
dark to light, represents the fault gouge, the high damage zone, and the low damage zone, respectively.
The black dashed lines represent the varying trends.
**Figure 5**. Major element comparisons over $TiO_2$ wt % to evaluate relative mobility during alteration.
Circles represent samples from wall rocks and damage zones, and triangles represent fault gouge. (a)
$TiO_2$ vs. $Al_2O_3$; (b) $TiO_2$ vs. $Na_2O$;(c) $TiO_2$ vs. $Fe_2O_3$; (d) $TiO_2$ vs. $K_2O$; (e) $TiO_2$ vs. MgO; (f) $TiO_2$ vs.
FeO; (g) $TiO_2$ vs. CaO; (h) $TiO_2$ vs. $P_2O_5$; (i) $TiO_2$ vs. MgO; (j) $TiO_2$ vs. LOI.
**Figure 6**. (a) A normalized Isocon diagram for the SB outcrop using the normalization solution. The
thick line indicates the unified isocon defined by $TiO_2$; the numbers before the oxide symbol represent
the scaling coefficients; and (b) Schematic illustration of mass changes in a three-sample system.
**Figure 7**. (a) $Na_2O$ concentrations vs. feldspar minerals concentrations; and (b) CaO concentrations
vs. carbonate minerals concentrations of wall rock, high damage zone, low damage zone and fault
gouge at the SB outcrop.
**Figure 8.** (a) $Al_2O_3$ concentrations vs. $K_2O$ concentrations of wall rock, high damage zone, low
damage zone and fault gouge at the SB outcrop. Ⅰ represents fault gouge and high damage zone
samples and most of low damage zone sample underwent illitization;Ⅱ represents wall rock samples
and one of low damage zone sample which did not undergo illitization. (b) illite concentrations vs. I/S
layer concentrations of wall rock, high damage zone, low damage zone and fault gouge at the SB





outcrop.
**Figure 9.** Microphotographs of the rock units. All samples are from fault gouge. (a) carbonate minerals
which altered to some clay minerals. (b), (c), (d), (e) and (f) feldspar and other minerals which altered
to chlorite. (b), (c), (d) and (e) images are presented in plane polarized light, and (a) and (f) image is
present in crossed polarized light.
**Figure 10.** EDS point analysis of feldspar particles in the middle location and the edge location. (a)
and (b) are the microphotographs of the rock units and the element contents for the middle and the
edge point, respectively.
**Figure 11**. EDS line scanning of the clay minerals formed by the alteration of feldspar particles. (a)
microphotographs of the rock units; (b) distribution of the main element contents, (c) distribution of
the Fe contents, (d) distribution of the Mg contents, (e) distribution of the Si contents.
**Figure 12**. The conceptual model showing geochemical and geophysical processes of faults in the
seismic period.






**Table 1**

| Major element | Fault gouge (Five samples) | High damage zone (Four samples) | Low damaged zone (Four samples) | Wall rocks (Two samples) |
|---|---|---|---|---|
| $SiO_2$ | 60.07 | 61.23 | 61.08 | 59.77 |
| $Al_2O_3$ | 17.45 | 14.33 | 12.99 | 11.87 |
| $Fe_2O_3^T$ | 6.37 | 5.10 | 4.70 | 3.73 |
| MgO | 2.44 | 2.95 | 2.61 | 1.95 |
| CaO | 2.03 | 2.72 | 4.45 | 7.77 |
| $Na_2O$ | 0.59 | 2.44 | 2.86 | 3.69 |
| $K_2O$ | 3.90 | 2.84 | 2.22 | 1.59 |
| MnO | 0.08 | 0.05 | 0.08 | 0.11 |
| $TiO_2$ | 0.71 | 0.68 | 0.61 | 0.49 |
| $P_2O_5$ | 0.19 | 0.21 | 0.22 | 0.23 |
| LOI | 5.72 | 6.92 | 7.76 | 8.25 |
| FeO | 3.80 | 2.82 | 2.25 | 1.86 |

* Major element data are in wt.%. Major element contents are the average values.





**Table 2**

| Sample | Material balance equation | M% |
|---|---|---|
| **SB-1-7** | 100 g wall rock – 15.00 g $SiO_2$ – 1.61 g $Al_2O_3$ – 5.34 g CaO – 2.54 g $Na_2O$ – 0.07 g $MnO_2$ – 3.72 g LOI – 0.07 g $P_2O_5$ → 72.74 g gouge + 0.02 g $Fe_2O_3^T$ + 0.53 g MgO +0.54 g K2O | 27.26 |
| **SB-3-1** | 100 g wall rock – 17.54 g $SiO_2$ – 0.13 g $Al_2O_3$ – 4.68 g CaO – 3.53 g $Na_2O$ – 0.03 g $MnO_2$ – 3.33 g LOI – 0.14 g $P_2O_5$ → 72.24 g gouge + 0.50 g $Fe_2O_3^T$ + 0.06 g MgO + 1.09 g K2O | 27.76 |
| **SB-1-8** | 100 g wall rock – 19.32 g $SiO_2$ – 6.94 g CaO – 3.59 g $Na_2O$ – 0.04 g $MnO_2$ – 5.13 g LOI – 0.15 g $P_2O_5$ → 68.38 g gouge + 0.89 g $Fe_2O_3^T$ + 1.17 g $Al_2O_3$ + 0.19 g MgO + 1.31g K2O | 31.62 |
| **SB-1-9** | 100 g wall rock – 20.02 g $SiO_2$ – 6.92 g CaO – 3.59 g $Na_2O$ – 0.05 g $MnO_2$ – 5.13 g LOI – 0.17 g $P_2O_5$ → 68.38 g gouge + 1.30 g $Fe_2O_3^T$ + 0.8 g $Al_2O_3$+ 0.19 g MgO + 1.27 g K2O | 31.67 |
| **SB-2-1** | 100 g wall rock – 17.08 g $SiO_2$ – 6.84 g CaO – 3.57 g $Na_2O$ – 0.05 g $MnO_2$ – 5.13 g LOI – 0.16 g $P_2O_5$ → 67.31g gouge + 0.82 g $Fe_2O_3^T$ + 0.59 g $Al_2O_3$ + 0.19 g MgO + 1.18g K2O | 32.69 |
| **SB-4-1** | 100 g wall rock – 13.47 g $SiO_2$ – 0.11 g $Al_2O_3$ – 0.32 g $Fe_2O_3^T$ – 6.21 g CaO – 3.50 g $Na_2O$ – 0.04 g $MnO_2$ – 5.13 g LOI →70.2 g gouge + 0.19 g MgO + 1.00 g K2O + 0.12 g $P_2O_5$ | 29.8 |







**Table 3**

| Fault zone subdivision (sample numbers) | Material balance equation | M% |
|---|---|---|
| **Fault gouge** (5) | 100 g wall rock – 18.11 g $SiO_2$ – 0.26 g MgO – 6.36 g CaO – 3.27 g $Na_2O$ – 0.05 g $MnO_2$ – 4.28 g LOI – 0.10 g $P_2O_5$ → 69.63 g gouge + 0.24 g Al2O3 + 0.69 g $Fe_2O_3^T$ + 1.12 g $K_2O$ | 30.37 |
| **High damage zone** (4) | 100 g wall rock – 15.32 g $SiO_2$ – 1.46 g $Al_2O_3$ – 0.02 g $Fe_2O_3^T$ – 5.80 g CaO – 1.91g $Na_2O$ – 0.07 g $MnO_2$ – 3.22 g LOI – 0.08 g $P_2O_5$ → 72.78 g gouge + 0.19 g MgO + 0.48g $K_2O$ | 27.22 |
| **Low damage zone** (4) | 100 g wall rock – 10.07 g $SiO_2$ – 1.29 g $Al_2O_3$ – 0.02 g $Fe_2O_3^T$ – 4.15 g CaO – 1.36 g $Na_2O$ – 0.04 g $MnO_2$ – 1.93 g LOI – 0.05 g $P_2O_5$ → 72.78 g gouge + 0.18 g MgO + 0.22 g $K_2O$ | 18.40 |





**Figure 1**








**Figure 2**


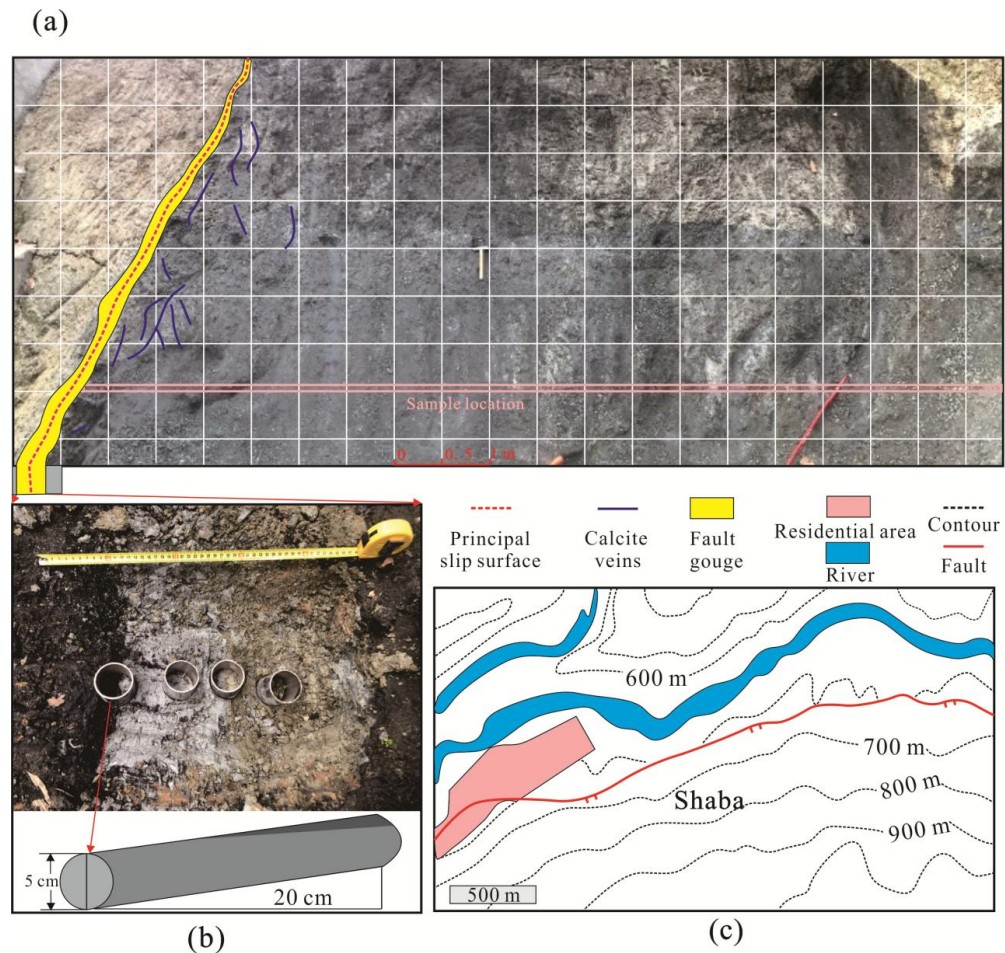





**Figure 3**

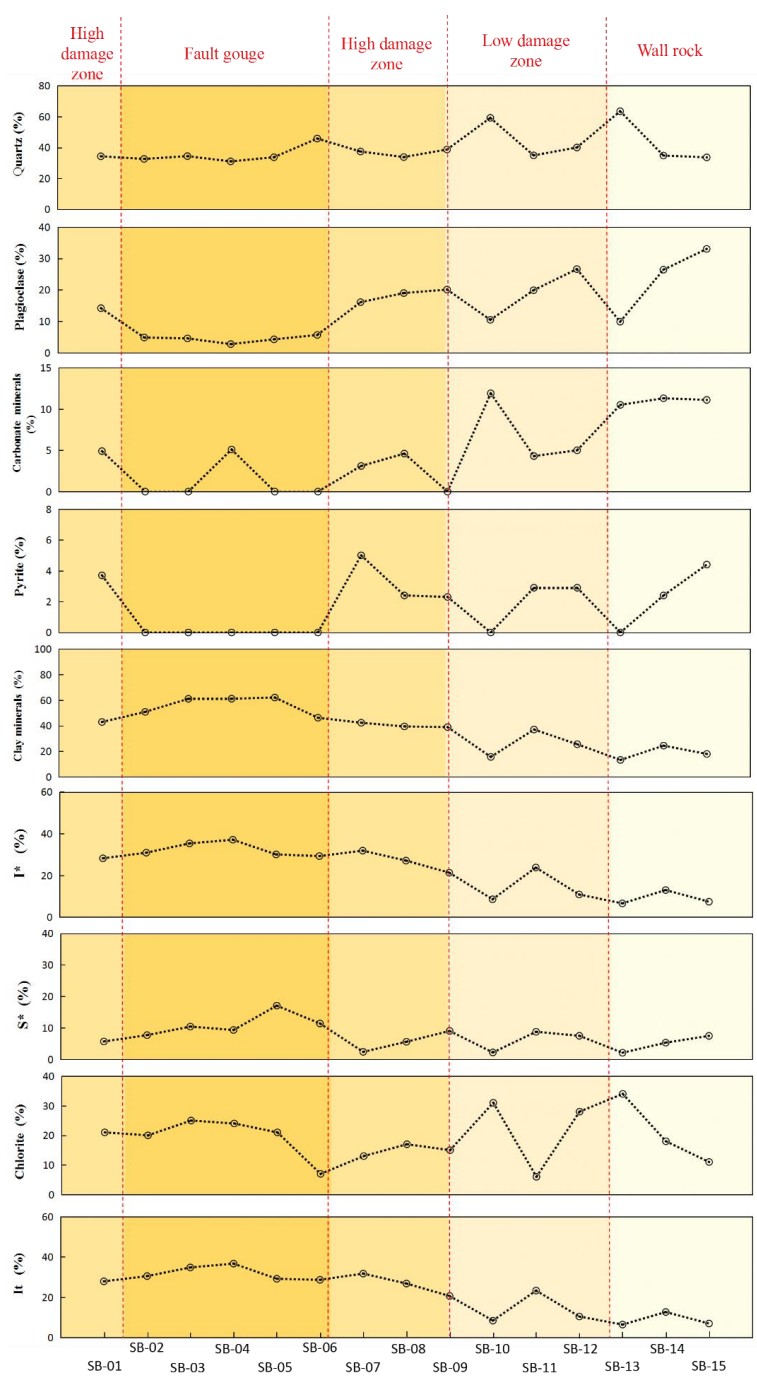





**Figure 4**

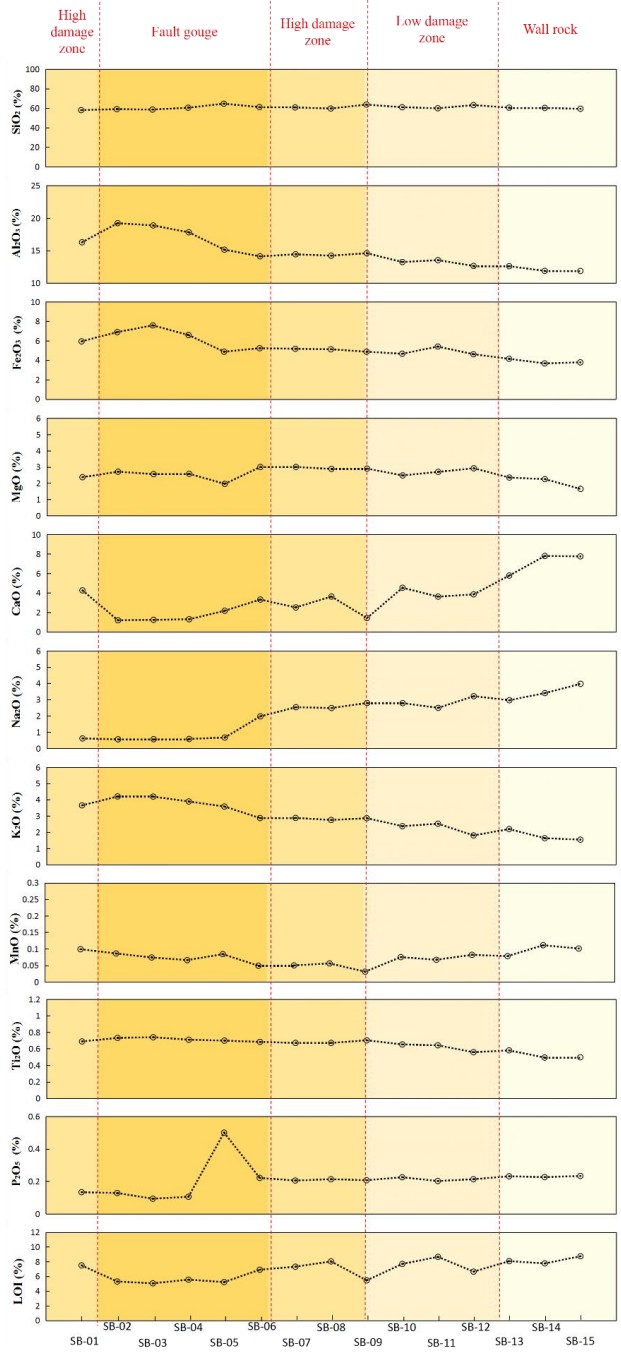





**Figure 5**



Fault gouge sample   Non-fault gouge sample






**Figure 6**

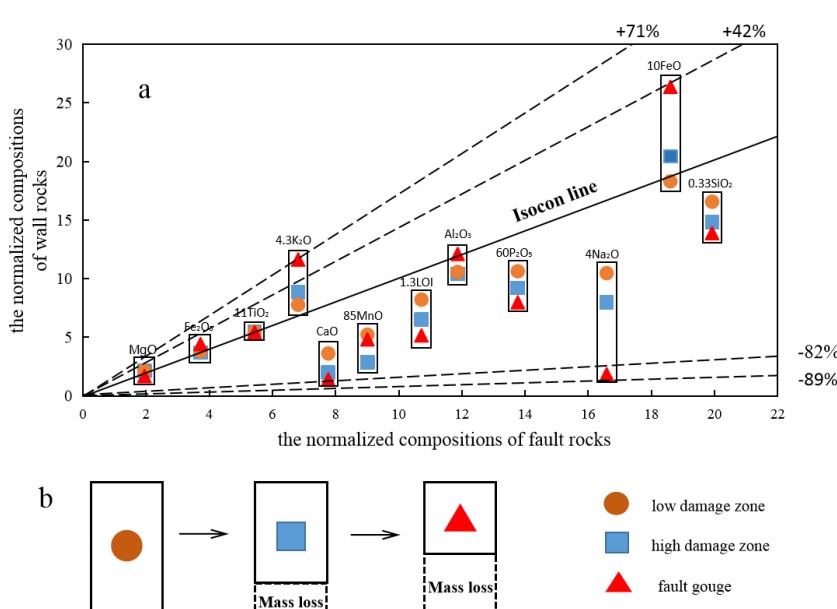







**Figure 7**

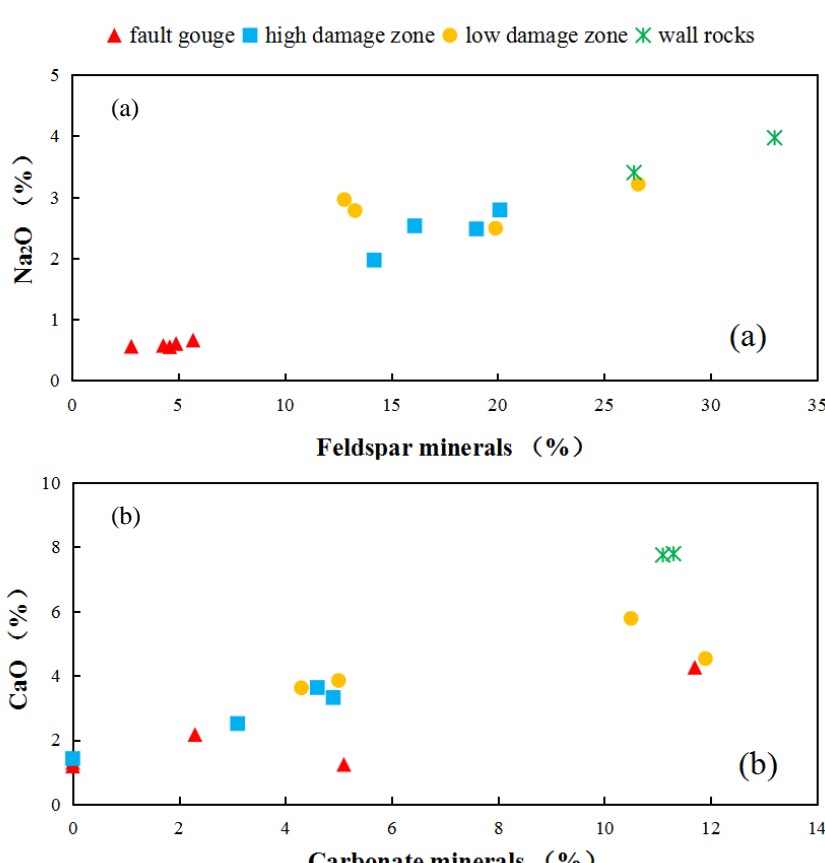






**Figure 8**

(a)

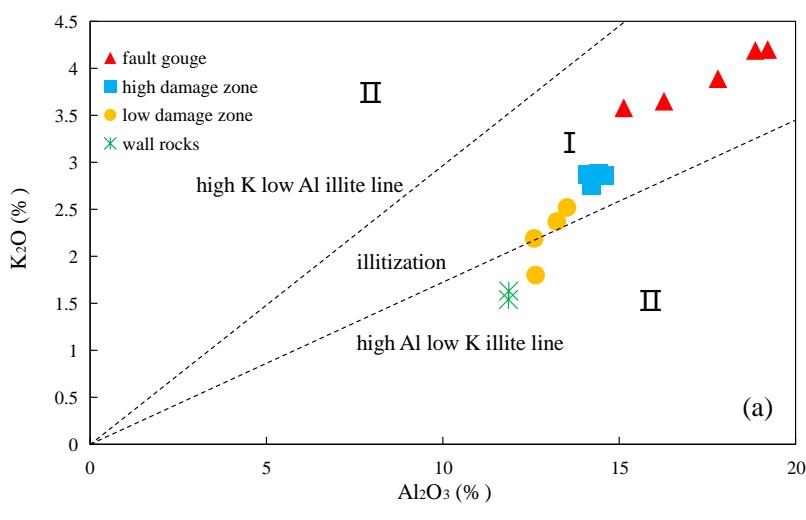

(b)

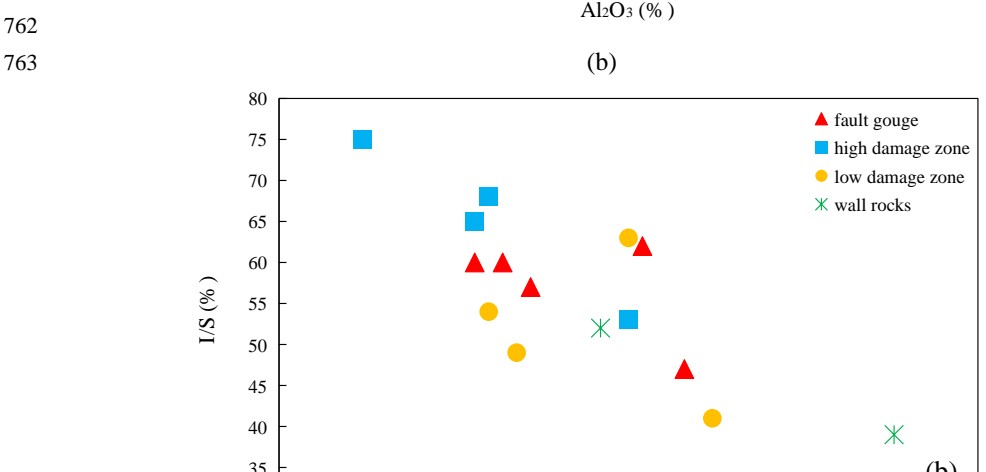



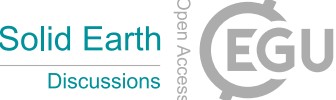

**Figure 9**
(a)                                   (b)

(c)                                   (d)

(e)                                   (f)



**Figure 10**

(a)

| Element | Concentration (%) | | | Chemical component |
|---------|---------|------|----------|--------------------|
|         | quality | atom | compound |                    |
| Mg      | 1.19    | 1.05 | 1.97     | MgO                |
| Al      | 13.40   | 10.68| 25.31    | Al$_2$O$_3$        |
| Si      | 26.51   | 20.30| 56.71    | SiO$_2$            |
| K       | 6.79    | 3.73 | 8.18     | K$_2$O             |
| Fe      | 6.08    | 2.34 | 7.83     | FeO                |
| O       | 46.03   | 61.89|          |                    |
| Total   | 100.00  |      |          |                    |

(b)

| Element | Concentration (%) | | | Chemical component |
|---------|---------|------|----------|--------------------|
|         | quality | atom | compound |                    |
| Mg      | 4.56    | 4.55 | 7.56     | MgO                |
| Al      | 11.21   | 10.09| 21.18    | Al2O3              |
| Si      | 15.22   | 13.17| 32.56    | SiO2               |
| Fe      | 30.08   | 13.08| 38.70    | FeO                |
| O       | 38.93   | 59.11|          |                    |
| Total   | 100.00  |      |          |                    |





**Figure 11**



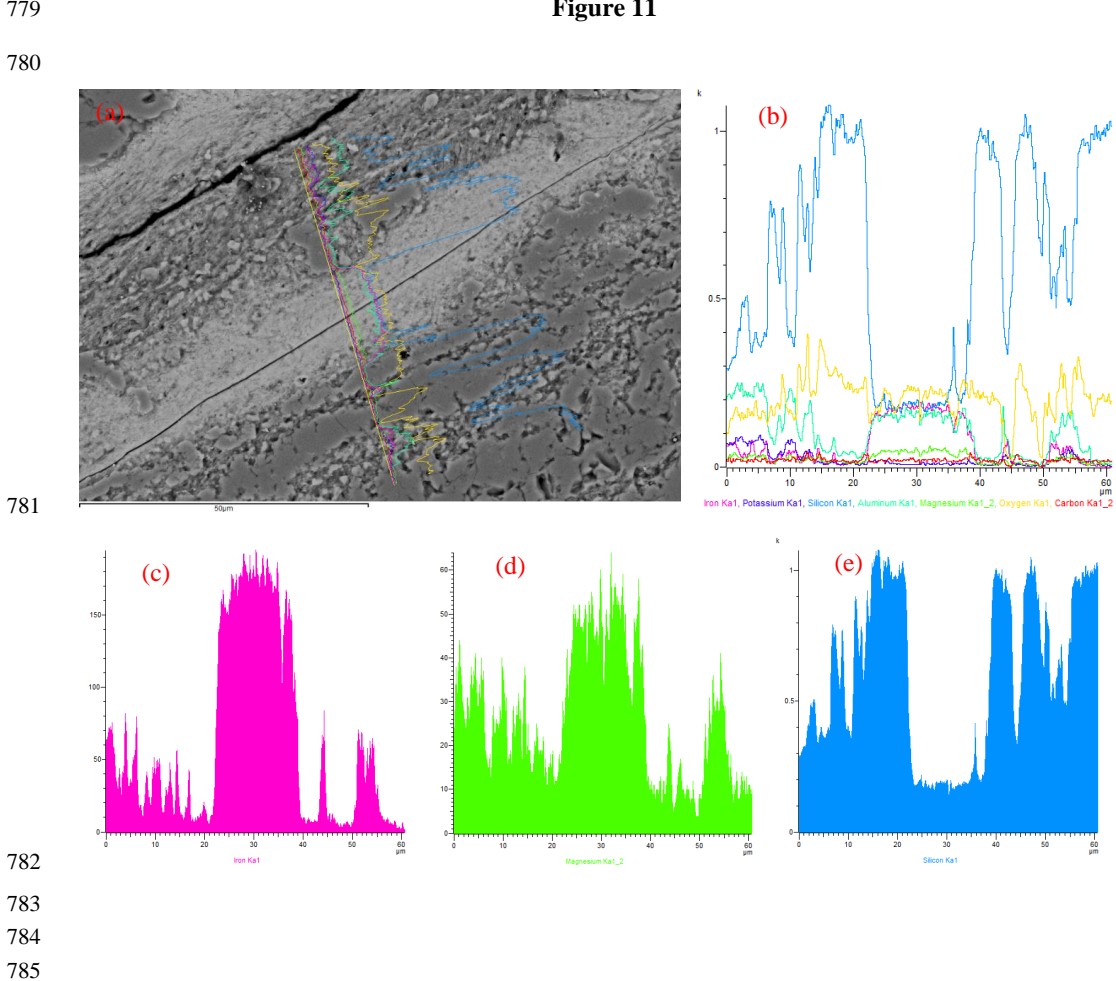











**Figure 12**

clastic sedimentary layer

The reduction in the grain size + decomposition of feldspar+ the formation and oriented arrangement of clay minerals

chloritization + precipitation of hydrothermal minerals + alteration of feldspar + the formation of calcite veins
