# Peer review of "Coseismic fluid–rock interactions in the Beichuan-Yingxiu surface rupture zone of the Mw 7.9 Wenchuan earthquake and its implication for the fault zone transformation"

_Solid Earth, 2020_

## Referee Comment (RC1) · Anonymous Referee #1 · 15 Sep 2020

The submitted manuscript reports on the mineralogical and geochemical characteristics of the surface rupture zone of the Mw7.9 Wenchuan earthquake. Based on the results, the authors discuss mineral reactions and mass transport processes along and or across the fault zone during coseismic as well as post-seismic periods. Generally, the manuscript is well written, and provides some important information about the evolution of this fault zone. While the methods and results of geochemical data and isocon analysis seem to be clearly shown, I have some concerns about the mineralogical data. The illitization in I/S and chloritization in the fault gouge are invoked as bases

for inference of coseismic heating (thermal pressurization). In my view, the authors should provide, not only the % values as summarized in Fig. 4, but also more information such as analytical procedures and raw XRD patterns. First of all, I'm confused about the sample preparation (Lines 147-153). In general, centrifugation is a process to separate the particles below $2\mu$ size (clay separates), but why did the authors grind the samples below $2\mu$ grain size before centrifugation? Regarding %S (or %I) in I/S mixed layer (Lines 147-153), the authors should provide details how to obtain the numbers, because there is often some difficulty in analyzing such property from XRD. Also, typical XRD patterns (at least in supporting information) would be helpful to prove the validity of the results. The authors discuss the reason for "extensive chloritization" in the fault zone (section 5.2.4 and conclusion 4), but the chlorite content in the host rock is highly variable, and one of the samples shows the higher (highest) value than in the fault zone. I'm wondering whether the chloritization is actually associated with faulting process. The additional information on the occurrence and chemical composition of chlorite in the host rock is necessary for this argument. It is also unclear why the LOI of the fault gouge shows the lower value than the others despite abundant clay minerals. In Line 141 the authors state "which suggests that the fault gouge has a relatively low water content and its fluid permeability is lower than that of the damage zone", does this mean the XRF was performed on wet samples? If so, the authors should describe how the samples were collected and stored to keep the wet states in 3. Sampling and experimental procedures section. Also, it may be better to check by calculation of frictional heating whether the temperature actually increases to the extent enough to trigger dehydration and clay transformation reactions under such shallow (i.e., low stress) and wide heating-zone ($\sim$40cm; Fig. 2) conditions.
* * *

---

## Referee Comment (RC2) · Anonymous Referee #2 · 25 Sep 2020

Review of Wang et al., SE-2020-117 The manuscript by Wang et al describes a study of fault rocks formed in the near-surface of the Beichuan-Yingxiu fault, which produced a Mw 7.9 earthquake in 2008. The central claims of the manuscript, as I see them, are as follows: 1) that the mineralogy and geochemistry of the fault rocks varies systematically across the identified architectural elements (i.e. gouge, damage zones, protolith); 2) that the patterns of mineral transformation and apparent mass loss in the fault core/main gouge zone are driven by coseismic frictional heating and thermal pressurization; and 3) that the patterns observed in the surrounding damage zones are

the result of post-seismic ingress of both surface derived and hydrothermal fluids. Of these, I find that only claim (1) is supported by the provided data. The remaining claims are entirely unsupported. More troubling, it is my opinion that claims (2) and (3) are effectively assumed by the manuscript, and the data are then interpreted selectively to support them. It is my recommendation that this manuscript be rejected without additional consideration.

The quality of the presentation of this manuscript also leaves much to be desired. I understand and sympathize that the authors are likely not native English speakers, but that does not reduce the requirement that for a work to be publishable, it must first be understandable. I struggled greatly in trying to understand many of the key portions of this manuscript as written. Some of these areas are noted below with suggestions for improvement, but this is far from an exhaustive list.

The most significant issue with this manuscript, in my opinion, is in the sheer number of assumptions regarding fault behavior. Yes, this fault has produced surface-rupturing earthquakes, so coseismic deformation must play a role on some level, but that is only necessarily true for deformation. It is not an a priori requirement that any of the mineral transformations, mass loss, other fluid-rock interaction occurred coseismically, with or without frictional heating. Independent evidence for these things needs to be presented. I would argue that all of the observations presented by the manuscript could just as easily (and perhaps more parsimoniously) be interpreted to be the result of "passive" fluid-rock interaction occurring entirely within the interseismic period. I do not recall a single line of presented evidence that would indicate frictional heating, thermal pressurization, thermal decarbonation, or any of the other processes argued by the authors to control fault-rock formation.

The data, although somewhat poorly described, are interesting, and therefore have the potential to be published someday. I would suggest that the authors focus on constructing a new manuscript, which simply presents the available data set clearly, and in a manner that is untainted by a preconceived narrative. Then, discuss the potential

mechanisms that could produce the observed trends. This would include significant discussion of the possibility that all of the observations are the result of passive fluid-rock interactions during the interseismic periods, unless independent, positive evidence can be produced to argue for a coseismic origin specifically.

Line Referenced Comments: 20-37: The role of fluids in influencing fault-rock mechanical and geochemical properties isn't at all debatable in my opinion. Fluids are a major factor, even if an incompletely understood one. The abstract overall is a bit convoluted and difficult to read. Part of this is due to the use of terms that are apparently specific to the studied fault zone (e.g. "upper and lower" damage zones, gouge "central-strong deformation region", etc). Finally, the ending sentence simply states that frictional heating, fracturing, and fluid-rock interaction affect the composition and mechanics of fault zones. Is this really the only new information provided by this work?

41: Fluid "action" cannot be "present", but fluids themselves certainly can be.

42: This definition of thermal pressurization is redundant. Sentence could be greatly simplified.

47: Suggest "Coseismic frictional heating may intensify fluid-rock interactions…", or similar. Your definition of fluid-rock interaction neglects the formation of cements/veins, sinters, pressure solution, etc which are major controls on fault rock hydromechanical properties. Suggest that devolatilization may be a better term than "deaeration".

57: Just macroscopically? Authigenic phyllosilicates are often nanometric in scale. This differentiation between macro and microscopic processes is kind of trivial. Both mineral alteration and geochemical enrichment/depletion occur at a variety of scales.

62: "Fluid" is misspelled in this line.

65-66: Probably true, but it would be useful to state how so here.

66-68: Again, I don't think the role of fluid-rock interaction in fault-zone development is debatable at all. Especially in sandstones, where a huge amount of work has been

conducted from the early 1990's until today. See work by Evans, Goodwin, Shipton, Williams, Fossen, Soliva, Balsamo, Storti, Eichhubl, Laubach, Mozley, and Petrie just to name a few.

68-72: This line states that a specific earthquake had never occurred before it occurred, which is of course true. Suggest rephrasing for clarity.

76-77: Fluid is misspelled again. I do not understand the differentiation that is being proposed here.

77-87: If I am honest, I cannot understand exactly what the authors are attempting to convey here. Based on what I can understand, it seems that most of this needs some corresponding citations. The questions at the end of this section are effectively "begging the question". The manuscript has not yet stated that these things have occurred in the exposed portions of the fault in order for the reader to wonder what their mechanisms and distribution may be.

88: I do not think an acronym is really required to describe one short word.

96: Same misspelling of "fluid".

104: Not sure what exactly "fresh" means in this case.

99-127: Section needs major rewriting for clarity. The only information I pulled out of this is that the fault is transtensional / oblique-slip normal with an apparent displacement magnitude on the order of 10 or so meters.

132: Figure 2 does not show/label any of the structural features described here other than the gouge layer, which is itself entirely obscured by the annotation.

150-151: How were they deposited? Smear, vacuum, gravity settling?

159: You have not defined reference intensity ratio (RIR). This method relies on "spiking" the sample with a known mass of corundum powder. Was this done? If so I do not see where that is described.

162-167: I do not think it is reasonable to push all of your XRF methodology descriptions to other sources. A summary here would be useful.

170: So you sieved out the >2 mm grains prior to grinding for xrd? Why?

173: Why not just specify what clay minerals were there rather than just one that wasn't?

174-175: What is this line trying to say?

183: The matrix?

193: Previously you said there was no smectite. Now there is illite/smectite, which is typically classified as a smectite.

216-220: I'm sorry, but a) where does fracturing come into this? b) how do we know necessarily that any of this is coseismic? c) what does an "open dynamic geological process" mean?

256: Without time, this cannot be a "rate".

289: Again, how do we know this is coseismic?

311: Again, not a rate.

325-327: The two clauses in this sentence effectively state the same thing.

335-344: Ok, so frictional heating could be a mechanism of transforming smectite to illite. Any direct evidence that it actually was? That transition does occur in the crust in the absence of frictional heating.

345-348: Seems like some citations for this assertion would be appropriate.

353-354: Or, it was just dissolved during fluid-rock interaction in an acidic environment, which you argued was the case locally in the last paragraph. The fact that the loss of carbonate minerals extends beyond the principal slip surface into the surrounding damage zone is yet more evidence that the process is driven by dissolution rather

than thermal decarbonization. Peak frictional heating temperatures on faults dissipate rapidly to low values in the surrounding rock.

389-390: With some exceptions, we typically think of chlorite authigenesis occurring somewhere in the temperature range of 150-200 C or greater. So, it would seem that this manuscript is invoking very hot fluids within the uppermost few meters of the surface? Seems this would preserve some evidence of boiling?

413-415: I have yet to see a single line of evidence indicating that the mineral transformation and elemental mass change occur coseismically or even shortly thereafter. Not one. Why is this narrative being pushed so hard? Why not simply describe the system, and discuss the potential ways in which it may have developed. This section effectively assumes the "answer" and interprets the data selectively in a way that fits the preferred narrative. That is not good science.

---

## Referee Comment (RC3) · Catriona Menzies (Referee) · 28 Sep 2020

Wang et al report on alteration of fault rocks due to co- and post-seismic fluid-rock interactions and provide XRD data confirming mineralogical changes and XRF data illustrating chemical changes, mostly of major elements. The manuscript requires some significant work to put it into context, fully use the data they present as evidence for their findings, discuss their data and findings with reference to other studies of similar processes to bring the work up to publishable level. Currently there is significant assumption throughout the discussion that coseismic heating and thermal pressurisation

drive the changes/ variation they report with no investigation of alternative processes, nor the use of their data to provide succinct evidence of these processes, this is clearly a significant flaw and renders the manuscript unpublishable in its current form.

From the outset, the English in the abstract is poor, and throughout it does not improve significantly, often with the meanings of sentences flipped due to poor English/ grammar/ wrong words used. For example, "contents of some elements" (line 464); here I think they mean the elemental composition of the rock, but how it is written makes it read as if it is the contents of the actual elemental make up they are measuring and that is entirely wrong and misleading. There are numerous examples throughout and without prior knowledge of geochemistry/ alteration/ fluid-rock interaction the meaning would be misconstrued by readers.

The overall principles of the paper are there, focussing on the alteration of primary minerals within a fault zone relating to both coseismic and postseismic hydrothermal/ fluid-mediated reactions, but there are key steps of the scientific method missing. Before this manuscript can be published it needs:

- Writing tidied up; grammar and correct use of terminology elements, oxides, components, concentrations, etc, etc are mixed up throughout. LOI is referred to as a component of the rock - when it is a measurement of the volatile content of the rock before and after heating, which is made up of $H_2O$, $CO_2$, etc etc.

- Introduction needs to be improved; it requires an overhaul of the geological setting and putting into context with published work (not just of work in the geographical area/ same fault zone).

- Systematic descriptions need to be made clearly and linked in with figures better. Some more figures would help with this as well as a summary figure of all observations. These observations are crucial for understanding this system and drawing comparisons with others and should be a main focus as an output from this paper.

Main technicalities that require attention:

- There is confusion about smectite - is it present or not? Is it primary in the rock and becomes illitised? It is stated that the conditions are not favourable for its formation hydrothermally, but there is confusion within the writing that makes the role of smectite here confusing for the reader (me).

- Define the protolith rock better and show that it is the same protolith in the gouge and wider fault zone - this must be done before any mass change calculations can be made

- Mass gain/ loss and isocon analyses: Group elements in figures - ie by colour to different "generations" or types of alteration that you describe. Such alteration should be systematically described before you investigate mass changes. Careful of language regarding loss/ gain; enrichment/ depletion. It is the loss of elemental components from the system (oxides as presented) that you are quantifying. It would be good to show an isocon plot of gouge vs wider fault zone (representative) as it is likely the gouge formed from the wider fault zone and this would be interesting to test. Additionally, throughout the results and discussion mass loss is referred to on many occasions without any reference to possible volume loss which may be important when thinking about the mechanical stability of the system. You could do this by looking at the reactions driving the loss and relative densities of different phases involved if you do not have densities of the different rocks measured. However, fig. 6 indicates there is also some mass gain happening - this should be discussed in context with reactions and mass losses that are suggested to occur. Can you split LOI into H2O and CO2 roughly by assuming all CO2 is attached to Ca? This would be rough but an estimation (with error bars) would be nice, plotting LOI like this is not very meaningful and may be misleading - for example you state the gouge has less H2O than surrounding fault zone, despite it having a significantly higher proportion of clay - how can this be? I think here you are ascribing lower LOI to lower H2O, when in reality your documented breakown of calcite in the gouge means the lower LOI is due to loss of CO2, not H2O.

- Where is there evidence that the reactions and changes you note in the fault zone (in the near surface) are due to the most recent Wenchuan earthquake and not previous earthquakes when these rocks were at a deeper level? For example, the chloritisation reactions you describe would need to be at at least 150C which is very unlikely in the near surface where they have been sampled. Perhaps this was not the aim, but the way it is presented indicates that the changes you document are due to the recent Wenchuan earthquake in shallow subsurface. Please clear this up.

- Link breakdown of specific minerals to what you show in your mass loss/ gain analyses better - can you link this to relative timing based on your petrological observations? This can then help to develop and test your hypotheses on coseismic and postseismic reactions and the drivers for these reactions.

- Evidence needs to be presented for the coseismic reaction drivers - stated as thermal pressurisation and dehydration drives mass loss - 1. what is the evidence for either of these processes (you need to tie it all together) 2. how do these processes contribute to mass loss? What reactions do they facilitate and how? There is literature on this which should be investigated and discussed along with the discussion of these data. What are the possible alternative processes that may drive the changes you see?

- The current discussion of data needs to be improved, relating the presented data to the proposed mechanisms better as well as drawing on the vast literature to discuss these findings in context with different processes elsewhere. Only once this is done can conclusions about the effect of fluid-rock interactions and coseismic reactions on seismic slip be commented on. The final discussion section requires a complete rewrite to satisfy this.

This is by no means an exhaustive review, some considerable work is required before the finer details can be finessed, and attending to those in this review would be largely irrelevant given the substantial rewrite required.